# Mechanical Fractionation of Adipose Tissue—A Scoping Review of Procedures to Obtain Stromal Vascular Fraction

**DOI:** 10.3390/bioengineering10101175

**Published:** 2023-10-09

**Authors:** Jan Aart M. Schipper, Constance J. H. C. M. van Laarhoven, Rutger H. Schepers, A. Jorien Tuin, Marco C. Harmsen, Fred K. L. Spijkervet, Johan Jansma, Joris A. van Dongen

**Affiliations:** 1Department of Oral & Maxillofacial Surgery, University Medical Center Groningen, University of Groningen, 9713 Groningen, The Netherlands; 2Department of Plastic, Reconstructive and Hand Surgery, Catharina Ziekenhuis Eindhoven, 5623 Eindhoven, The Netherlands; 3Department of Pathology & Medical Biology, University Medical Center Groningen, University of Groningen, 9712 Groningen, The Netherlands; 4Department of Plastic, Reconstructive and Hand Surgery, University Medical Center Utrecht, Utrecht University, 3584 Utrecht, The Netherlands

**Keywords:** tissue stromal vascular fraction, mechanical fractionation, adipose tissue, mechanical dissociation, adipose-derived stromal cells

## Abstract

Clinical indications for adipose tissue therapy are expanding towards a regenerative-based approach. Adipose-derived stromal vascular fraction consists of extracellular matrix and all nonadipocyte cells such as connective tissue cells including fibroblasts, adipose-derived stromal cells (ASCs) and vascular cells. Tissue stromal vascular fraction (tSVF) is obtained by mechanical fractionation, forcing adipose tissue through a device with one or more small hole(s) or cutting blades between syringes. The aim of this scoping review was to assess the efficacy of mechanical fractionation procedures to obtain tSVF. In addition, we provide an overview of the clinical, that is, therapeutic, efficacy of tSVF isolated by mechanical fraction on skin rejuvenation, wound healing and osteoarthritis. Procedures to obtain tissue stromal vascular fraction using mechanical fractionation and their associated validation data were included for comparison. For clinical outcome comparison, both animal and human studies that reported results after tSVF injection were included. We categorized mechanical fractionation procedures into filtration (*n* = 4), centrifugation (*n* = 8), both filtration and centrifugation (*n* = 3) and other methods (*n* = 3). In total, 1465 patients and 410 animals were described in the included clinical studies. tSVF seems to have a more positive clinical outcome in diseases with a high proinflammatory character such as osteoarthritis or (disturbed) wound healing, in comparison with skin rejuvenation of aging skin. Isolation of tSVF is obtained by disruption of adipocytes and therefore volume is reduced. Procedures consisting of centrifugation prior to mechanical fractionation seem to be most effective in volume reduction and thus isolation of tSVF. tSVF injection seems to be especially beneficial in clinical applications such as osteoarthritis or wound healing. Clinical application of tSVF appeared to be independent of the preparation procedure, which indicates that current methods are highly versatile.

## 1. Introduction

Clinical indications for adipose tissue therapy are expanding [1]. Initially, fat grafting has been widely used to restore volume loss due to trauma, ablative surgery, aging or congenital defects [2]. Later, indications for grafting of fat or components of fat expanded to more regenerative-based approaches such as improving wound healing or aged skin as well as treating scars [3]. This shift was partly caused by the development of stromal vascular fraction (SVF) of adipose tissue [4]. SVF is the fraction of adipose tissue without adipocytes. Adipose-derived SVF consists of an extracellular matrix and all nonadipocyte cells such as connective tissue cells: fibroblasts, adipose-derived stromal cells (ASCs) and vascular cells [5]. In the past few years, SVF has demonstrated its regenerative action by stimulating angiogenesis and immunomodulation [6,7,8].

SVF can be obtained by enzymatic isolation or mechanical fractionation, which yield, respectively, cellular SVF (cSVF) and tissue SVF (tSVF). cSVF is a single-cell suspension devoid of both adipocytes and extracellular matrix [9]. Mechanical fractionation is performed by forcing adipose tissue through a device with one or more small hole(s) or cutting blades between syringes. The holes or blades disrupt adipocytes leaving only the SVF. Subsequent fractionation, often by centrifugation, separates adipose tissue in an upper oily fraction from disrupted adipocytes, a middle layer of stromal vascular fraction, a lower liquid fraction with infiltration fluid, and a pellet of cellular debris. Mechanical fractionation yields tSVF, in which the original tissue structure is maintained (including extracellular matrix) but fragmented. The extracellular matrix functions in vitro as a scaffold for cells and a release reservoir of already-bound paracrine factors to improve regenerative processes [10,11]. It is thought mechanical fractionation holds higher therapeutic potential than enzymatic isolation because of the additional advantages of the intact extracellular matrix because it is less time consuming to make and less expensive [5]. Moreover, enzymes used for isolation are legally forbidden for clinical application in many countries [12,13,14].

After fractionation of lipoaspirate, separation methods such as decantation, filtration or centrifugation are used to separate the different fractions of lipoaspirate. The first developed procedure was the nanofat procedure [15]. The nanofat procedure uses decantated lipoaspirate, forcing it through a device with one hole (a fractionator) multiple times between syringes, and then filtrating it to obtain processed lipoaspirate. Later on, other techniques were combined with centrifugation, for example, respectively, the fractionation of adipose tissue (FAT) procedure or Lipocube [16,17]. 

All these fractionation procedures have been reported in both in vitro and clinical studies. Meanwhile, the development of most mechanical fractionation procedures changes continuously through small procedural optimizations. Due to this abundance in procedures and associated publications, it is unclear what the differences between these mechanical fractionation procedures are, for which clinical applications these procedures can be used and whether certain procedures may be preferred over others for specific indications. Efficacy of these procedures is determined by efficient removal of adipocytes, which can be assessed by volume reduction and absence of (immuno)histochemical markers of adipocytes. Hence, this review is conducted to assess the efficacy of mechanical fractionation procedures to obtain tSVF. In addition, we provide an overview of the clinical, that is, therapeutic, efficacy of tSVF on skin rejuvenation, wound healing and osteoarthritis.

## 2. Procedures

In general, three steps can be distinguished in the production of tSVF (Table 1): (1) excessive liquids such as oil, serum and infiltration liquid have to be removed (in jargon: drying of fat); (2) adipocytes have to be adequately disrupted by mechanical fractionation, that is, dissociation through a device with holes or sharp blades (a fractionator) by intersyringe shuffling; (3) processed tissue has to be separated from the damaged adipocyte debris and their contents, oil. These procedures can be categorized into three groups based on their use of tissue separation methods: (1) filtration, (2) centrifugation or (3) a combination of filtration and centrifugation (Table 2). 

tSVF is characterized by (1) absence of adipocytes determined by (immuno)histochemical staining of adipocyte markers and volume reduction, (2) presence of extracellular matrix by (immune)histochemical staining and (3) tSVF composition based on expression of cell-specific cluster of differentiation (CD) surface markers (Table 3, Table 4 and Table 5).

(1) Mechanical fractionation of tSVF can only be obtained by disruption of adipocytes through a fractionator. Their contents that is triglycerides (oil) are released after disruption of the cell membrane. Adipose tissue consists in volume of approximately 90% adipocytes [30], although the volume of adipocytes also depends on donor adipocyte cell hypertrophy which is influenced by factors such as body mass index [31]. Mechanical fractionation should therefore result in a ratio of approximately 9:1 of oil:tSVF. Hence, when most of the oil from the adipocytes is removed, the volume of the end product that is tSVF should be 90% less than the starting volume of lipoaspirate. The end volume of this fraction should therefore be reported to properly assess the efficacy of a mechanical fractionation procedure.

(2) The presence of extracellular matrix can be confirmed using histochemical staining such as Masson’s Trichrome staining.

(3) In 2013, a joint statement of the International Federation of Adipose Therapeutics (IFATS) and International Society for Cellular Therapy (ISCT) was published regarding the characterization of cultured ASCs by phenotype using flow cytometry and by function using colony forming units as well as multilineage differentiation [9].

### 2.1. Filtration Procedures

In total, seven mechanical fractionation procedures are equivalent to the nanofat procedure [15,18,19,20,24] of which four are classified as filtration procedure using either a nylon cloth or a filter device [15,18,19,20]. However, nanofat 2.0 and nanofat cell aggregates are almost identical to the original nanofat procedure. Nanofat 2.0 lacks a last filtration step and nanofat cell aggregates use a metal mesh screen instead of a nylon cloth. LipocubeNANO is the other mechanical fractionation procedure that we categorized as a filtration-only procedure (without the use of centrifugation).

Three out of the four nanofat procedures classified as filtration procedure do not mention the end volume of the fraction of processed tSVF [15,18,20] or the number of adipocytes that are still present in the processed tSVF. The authors of nanofat cell aggregate reported a volume reduction of only 15%, while centrifugation methods such as the FAT and SVF gel procedures acquire a volume reduction of 90%. Moreover, almost no histological evaluation has been performed on tSVF isolated using nanofat procedures, that is, filtration procedures. Authors of nanofat 2.0 only reported microscopic images using an oil-red O staining showing a different tissue structure compared to lipoaspirate, although perilipin is the required cytoplasmatic staining to evaluate whether the adipocyte cell membrane is intact. Perilipin-positive cells would therefore suggest there are still intact adipocytes present. The authors of the original nanofat procedure only performed viability staining and showed that no viable adipocytes were present [15]. Due to the minimal amount of data available, it remains unclear whether small modifications to the original nanofat procedure as a filtration method resulted in improvements in the purity of tSVF.

In contrast to centrifugation, which separates the oily from the aqueous fraction, filtration retains part of the adipocytes and therefore reduces the efficiency of the generation of tSVF. A recent study found that filtration of lipoaspirate reduced the number of SVF cells and extracellular matrix content compared non-filtrated lipoaspirate [32]. No studies reported the measurement of extracellular matrix in the validation data of these filtration procedures during the nanofat procedure.

None of the filtration studies reported CFU-F assay and only three studies reported adipogenic differentiation capability [15,19,20], while at least three differentiation lineages are warranted. Only the study describing the nanofat 2.0 procedure reported an ASC phenotype after the third passage. SVF composition based on CD marker expression has been evaluated in multiple studies; however, only the authors of LipocubeNANO used combined CD marker expression to determine different cell populations [20]. Multiple CD markers, both positive and negative, are needed to define different cell populations because single CD markers are expressed by multiple cell types. It should be noted that in none of the studies did the authors use adipocyte-specific markers, yet these cells are readily scored by virtue of their size during FACS analyses. The use of too few distinct CD markers results in an over- or underestimated number of a certain cell type. SVF cells can be divided in two large cell populations: blood-derived (CD45+) and adipose-derived (CD45−). Adipose-derived populations can be divided into endothelial-like cell types (CD31+) and stromal cell types (CD31−). The authors of LipocubeNANO defined the CD45−/CD31+ as endothelial cells (11.99%), the CD45−/CD90+ as well as CD73+/CD90+ populations as ASCs (combined 45.21%) and the CD45+/CD14+ population as monocytes/macrophages (2.43%) [20]. The actual number of ASCs is probably overestimated because both CD73+ and CD90+ (aka Thy-1) are expressed by endothelial cells (CD90 only on activated ECs) as well, while CD90 is also abundantly expressed on other connective tissue cells such as fibroblasts. The estimation of the number of ASCs would be more accurate if CD31+ was added to CD73+/CD90+ to differentiate between endothelial cells (CD31+) and ASCs (CD31−). Of note, monocytes express CD31 too, albeit at lower levels than ECs.

### 2.2. Centrifugation Procedures

Mechanical fractionation was performed in four out of eight procedures using a Luer lock fractionator/sizing transfer [16,21,24] and in four procedures using a blade-like fractionator [17,22,23,25,33]. Centrifugation is a frequently used step in the fractionation of adipose tissue to obtain SVF. Seven procedures use centrifugation as a first separation step and six procedures use centrifugation as a final separation step [16,17,21,22,23,25,33]. Centrifugation as a final separation step results in several fractions: an upper oily layer, a middle tissue layer rich in cells and with an extracellular matrix, a lower liquid layer consisting of serum and infiltration fluid, and a pellet with debris and cells not bound to the extracellular matrix (Table 1). Different authors use different layers for their end-product: Lipocube and MEST use the pellet layer, but mechanical micronization—squeeze and the FAT procedure use the middle tissue layer. Centrifuge-modified nanofat and evo-modified nanofat use centrifugation prior to intersyringe processing and are therefore more comparable to the centrifugation procedures than the original nanofat procedure, which is classified as a filtration procedure [24].

Only four of eight studies reported the end volume (52% to 90% reduction) [16,21,22]. Volume reduction differs significantly between the types of fractionators. The authors of the FAT procedure used a Luer lock one-hole 1.4 mm or three-hole 1.4 mm fractionator and reduced the lipoaspirate by 90% in volume. Immunohistochemistry showed a reduced number of adipocytes using a perilipin staining, which resulted in a significant enrichment of small vessels and the extracellular matrix. Mechanical micronization—squeeze used an automated device with sharp rotating blades in a syringe and reduced lipoaspirate by 50% in volume. Sharp propellor-like blades in the piston spin electronically, moving up and down twice [22]. Centrifugation force was 956× *g* for 2.5 min for the FAT procedure and 1200× *g* for 3 min for the mechanical micronization procedure. While the FAT procedure resulted in tSVF that was virtually devoid of adipocytes [16], the authors confirmed the presence of adipocytes in the mechanical micronization—squeeze procedure with perilipin staining; however, no quantification was provided [22]. This large difference in tissue volume reduction indicates that the one- or three-hole fractionator of the FAT procedure more efficiently disrupts adipocytes than the blades of the mechanical micronization procedure. However, no quantification of adipocytes or extracellular matrix was provided. Authors describing the Lipocube and MEST procedures did not report the degree of volume reduction and used only the cell pellet after centrifugation [17,25]. This cell pellet contained 86–91% viable cells in Lipocube and 91–94% viable cells in the MEST procedure. The adjustable regenerative adipose tissue transfer (ARAT) histologically showed intact adipocytes; however, no quantification was provided [25]. Studies describing Lipocube, centrifuge-modified nanofat and evo-modified nanofat did not report any histological or immunohistochemical data. In the procedures without a final centrifugation step, that is, the centrifuge-modified nanofat procedure or evo-modified nanofat procedure, no end volume of the used fraction to assess the adequate removal of oil and debris was reported.

The authors of the FAT procedure showed immunohistochemical confirmation that an extracellular matrix was present in tSVF and showed that there was enrichment of the extracellular matrix compared to the control lipoaspirate [16,34]. No other procedures provided validation data regarding extracellular matrix.

Centrifuge-modified nanofat and evo-modified nanofat reported only cell yield (numbers) and immunophenotypic characterization using flow cytometry. It is therefore not clear if these modifications to the original nanofat procedure improved the original nanofat procedure. Differentiation capacity to prove the capability of cells to differentiate into osteogenic, adipogenic or chondrogenic cell lineage and the colony formation assay to assess clonogenic potential of these cells was only reported in the two FAT procedures [16,21]. Cultured ASC phenotype was reported by the FAT procedure after 2–4 passages [16]. All other procedures reported SVF composition [16,17,21,22,23,24,25]. SVF composition based on multiple CD marker expression was reported for only two procedures: tSVF obtained using the FAT procedure was composed of the following cell populations: CD45−/CD90+/CD105+ for ASCs (41.4%), CD34+/-CD31–/CD146+ for pericytes (0.3%), CD31+/CD34+ for endothelial cells (12.0%), CD45+/D34− for leukocytes (5.3%) and CD45+/CD34+ for hematopoietic stem cell-like cells (0.1%) [21]. The amount of ASCs/pericytes in tSVF obtained using the FAT procedure is comparable to LipocubeNANO. Although multiple cell populations are determined, almost 40% of the cell types within isolated tSVF remains unknown. tSVF derived from the Lipocube showed the following subpopulations: CD45−/CD90+ as well as CD73+/CD90+ for ASCs (combined 94.45%), CD45−/CD31− for endothelial cells (21.06%) and CD45+/CD14+ for macrophages/monocytes (7.28%). The total percentage of distinct types of cells in tSVF obtained using Lipocube is 122.79%, which is impossible [17]. Hence, according to these data, 94.45% of the cells in tSVF would account for ASCs, which is a gross overestimation.

### 2.3. Procedures Using a Combination of Centrifugation and Filtration

Three procedures used a combination of centrifugation and filtration: SVF gel, mechanical micronization emulsification and supercharge-modified nanofat [22,24,26]. SVF gel and mechanical micronization emulsification both use a first centrifugation step and as a final step both centrifugation and filtration [22,26].

SVF gel reduced 90% in volume, while mechanical micronization emulsification reduced 61% in volume when measuring the tissue in the filter and 90% when measuring the fluid after filtration [22,26]. However, the fluid after filtration in the mechanical micronization procedure consisted of extracellular matrix fragments, very few intact adipocytes and many dead cells (viability 39.3% ± 9.1). Supercharge-modified nanofat uses both a mesh filter and centrifugation in which the pellet of the flowthrough is collected and then added to mechanically processed tissue [24]. The diameter of the hole of the fractionator was not described in supercharge-modified nanofat. The modified nanofat procedures did not report the percentage of volume reduction. Confocal laser scanning microscopy showed that after the SVF gel procedure, only a few adipocytes were present while capillaries were fragmented but the density of vessel-associated connective tissue had increased. This was a logical consequence of near-complete removal of adipocytes. The FAT procedure is quite similar to the SVF gel procedure; the only difference is an extra filtration step in the SVF gel procedure. The SVF obtained after either procedure showed an enrichment in vasculature while presence of adipocytes was reduced to a bare minimum. The authors showed that the mechanical micronization emulsification procedure resulted in irregular-sized and irregular-shaped adipocytes and fragmented capillaries in the tissue before filtration, but that the fluid resulting after filtration showed few live cells, cell remnants and ECM fragments (viability 39%). Studies on supercharge-modified nanofat did not report (immuno)histology or end volume of the product.

The authors of mechanical micronization emulsification and SVF gel confirmed the presence of the extracellular matrix using scanning electron microscopy. Supercharge-modified nanofat did not test for extracellular matrix content.

According to the minimal definitions of the IFATS statement, only SVF gel showed differentiation capability [26]. SVF gel and mechanical micronization emulsification reported SVF composition, but did not report quantitative data [22,26]. Authors of supercharge-modified nanofat reported SVF composition [24]. There were no studies that reported CFU assays or cultured ASC phenotype.

### 2.4. Studies Describing Direct Comparisons between Procedures

Osinga et al. studied the effects of intersyringe processing with a three-way stopcock with 2 mm diameter for each hole [35]. This study shows that tissue viability was not affected by intersyringe processing 1, 5 or 30 times. Although, other studies show that a smaller diameter of the fractionator increases shear stress on tissue during processing, which promotes adipocyte disruption resulting in oil formation [36,37]. To isolate and condense SVF instead of emulsifying adipose tissue, an additional step of centrifugation prior to fractionation is necessary [38]. A comparison study between nanofat and SVF gel showed that volume is reduced 20% using the nanofat procedure and 80% in the SVF gel procedure [39]. The number of viable SVF cells was also 10-fold higher in SVF gel compared to nanofat. The increase in concentration of cells and volume reduction can be explained by the fact that nanofat lacks two centrifugation steps. The first centrifugation step (instead of decantation of the nanofat procedure) is necessary to remove excess liquid from the adipose tissue to ensure adequate disruption of adipocytes upon usage of the disruptor mounted on syringes [38]. The second centrifugation step is essential to separate oil from tSVF, as described above earlier. Furthermore, the combination of centrifugation and fractionation adds additional mechanical shear stress to SVF cells in comparison with fractionation alone. Pallua et al. showed an increased concentration of SVF-associated cells: ASCs (CD31−, CD34+, CD45−) and endothelial cells (CD31+, CD34+, CD45−) in the FAT procedure (two times centrifugation) compared to the nanofat procedure (no centrifugation) [40]. bFGF, IGF-1, PDGFBB, VEGF-A and MMP-9 expression levels were similar between the FAT procedure and the nanofat procedure. Banyard et al. confirmed that exposure to mechanical shear stress by intersyringe processing of the nanofat procedure results in cells with a higher expression of multiple non-SVF markers in processed lipoaspirate compared to standard unprocessed lipoaspirate: CD34 (threefold), CD13 (threefold), CD73 (twofold), CD146 (twofold), CD45 (twofold) and CD31 (twofold) [41].

## 3. Clinical Applications

As mechanical fractionation methods using intersyringe processing are convenient and fast, these are broadly applied and also modified by various user groups. Therefore, the variation in study and application areas is also continuously expanding [1]. Only research with mechanical fractionation methods using intersyringe processing to provide tSVF as stated above, and use of validated tools in analysis of clinical efficacy post-injection was discussed for the current overview. Therefore, lipogems was not included, as this method uses a device with metal balls inside that is manually shaken [42]. In addition, MyStem was excluded, since it did not describe which type of connector is used for processing [43,44].

To improve readability, we categorized the literature in the following (pre)clinical applications: skin and volume enhancement, wound healing, osteoarthritis and others.

In total, 45 articles (29 human, 16 animal) were eligible according to our search criteria in which tSVF was mechanically isolated and clinical outcome was reported with validated tools or histopathological analysis. Four RCTs (14%, 4/29) were conducted, and the majority of articles was prospective (91%). In total, 1465 patients (median follow-up 12 months (range 0.5–12); Table 6 and 410 animals were studied (median follow-up 3 months (range 0.5–6); Table 7). As study design and outcome measurements varied widely, no direct comparisons or meta-analysis of the included studies were performed.

### 3.1. Skin and Volume Enhancement

Sixteen studies reported the effect of tSVF on skin and volume enhancement in human study subjects, mainly in the facial area. Tissue SVF was obtained using the FAT procedure (*n* = 10), nanofat (*n* = 5) and *SVF gel* (*n* = 1). Most commonly, higher satisfaction and improvement in volume restoration were reported in study patients treated with tSVF (Table 6).

Fifty-six percent (9/16) of the studies included a control study group, of which only two (2/16, 13%) were conducted as RCT [45,46]. In three studies, tSVF was compared to Coleman’s fat, and all three showed significantly higher patient satisfaction in the tSVF-treated group [46,55,60]. One RCT (*n* = 63) also showed a higher retention rate (41.2 ± 8.4% vs. 32.6 ± 8.8%) and volume ratio (0.87 ± 0.02 to 0.89 ± 0.03) of the tSVF group compared to Coleman’s fat [46]. tSVF was also compared to treatments with Botox in improving horizontal neck wrinkles [47] or hyaluronic acid in improvement of wrinkles and skin texture [54], in which patient satisfaction was higher after treatment with tSVF. In five studies [45,54,55,56,57], tSVF was studied in combination with treatment with platelet-rich plasma (PRP), of which the effect of adding tSVF to PRP was only studied in one RCT. This study (*n* = 28) showed no difference of the combination of tSVF and PRP compared to PRP alone in ageing skin quality or patient satisfaction (VISIA and FACE-Q) in terms of aging skin rejuvenation [45].

In four studies [46,47,52,56], histopathological results from biopsies taken at the surgical site pre- and post-injection were reported (baseline vs. 1/6/12 months FU). Besides increased elastic and collagen density [47,56], a decrease in CD4+ T cell infiltration in the basal layer was observed when comparing pre- and post-injection samples [52], and improvement in dermal cellularity (mainly fibroblasts) and vascular density of the superficial layer [56].

A direct comparison of techniques was studied by Zhu et al. in patients treated with tSVF for periorbital volume and skin rejuvenation, which was defined as correction of fine wrinkles of the lower eyelid and infraorbital dark circles [59]. Significantly higher satisfaction and fewer reoperations (secondary tSVF injection needed for volume restoration) were observed for the group treated with tSVF using the FAT procedure instead of tSVF harvested using a nanofat procedure. Nevertheless, in both tSVF-treated groups, clinical improvement was reported.

Skin and volume enhancement was also studied in seven animal models (mice, rats, rabbits and guinea pigs; Table 7). In four studies, xenografts from humans were used in immunocompromised animals [74,75,76,78]. All seven studies compared tSVF treatment with a control group. Histological biopsies showed that viable adipocyte architecture and collagen accumulation were observed with use of tSVF. In addition, a mononuclear infiltrate—predominantly macrophages—was seen on the periphery of the fat graft near the blood vessels. Increased dermal thickness, higher collagen deposition and more fibroblasts were also observed in mice, and increased mRNA-expression of TGF-β1 and Smad2 (nonphospho) expression was seen [74]. MMP-2 and MMP-9 were significantly lower in tSVF-treated mice compared to controls [75]. Nanofat-treated mice also showed better cell survival and integrity: fewer vacuoles, less fibrosis and inflammation, and the capillary density measured with CD31+ vessels was significantly higher than controls (24.6 ± 4.72 nanofat vs. 10.4 ± 2.88 control) [76]. Volume retention was studied in twelve rabbit ears by applying autologous tSVF (SVF gel) with plasma-rich fibrin (PRF) versus tSVF alone. Histological biopsies showed no distinct effect, though tSVF and PRF combined showed a more ordered fashion of adipocytes and fibroblast distribution, which was lacking in tSVF monotherapy. This indicates PRF may induce formation of fat cells [79]. An et al. used minced rat fat, which was fractionated by intersyringe shuffling through a fractionator with three 1 mm diameter holes [27]. No post-fragmentation filtration or centrifugation was applied. From histological biopsies, the average collagen density was higher in tSVF-treated rats. Levels of PCNA and CD31, representing, respectively, proliferation and vascularization, were also higher.

### 3.2. Wound Healing

Ten human studies reported improvements in wound healing after applying tSVF, of which five included a control group, two were RCTs. Tissue SVF was obtained using the FAT procedure (*n* = 2), nanofat (*n* = 4), centrifuge-modified nanofat (*n* = 3) and SVF gel (*n* = 1). In four studies, tSVF was added to PRP (three studies) and Coleman’s fat (one study). In general, the majority of studies showed positive results in usage of tSVF, such as improvement in POSAS and wound healing (Table 6).

After mammoplasty, FAT procedure-derived tSVF improved wound healing compared to in-patient saline-treated controls in an RCT with 40 female patients. A lower POSAS score for tSVF was reported at six months, though after 12 months follow-up, differences between tSVF and placebo treatment were less distinct (FAT vs. placebo; patient 14.4 ± 7.6 vs. 15.3 ± 9.0; observer 14.5 ± 6.4 vs. 14.6 ± 8.8). In addition, similar collagen alignment, depth and width were shown, though more organized post- (scar tissue) than preoperatively (normal skin). Another study reported significantly lower postoperative POSAS and histological results in 20 facial scar patients treated with tSVF (FAT procedure) [63]. Histologically analyzed biopsies of the scar pre- and post-injection showed no difference in elastic fibers, though higher melanin average optical density of the basal cell layer was reported. In addition, development of sebaceous and sweat glands was seen, which were absent in the baseline biopsy of the scar tissue. An RCT performed in 50 burn patients treated with Coleman’s fat and topical tSVF (centrifuge-modified nanofat) showed faster healing, less contractures and less reoperations when compared to 50 patients treated with conventional silver sulphadiazine cream and split skin grafting therapy [62]. In addition, biopsies showed more rapid collagen deposition, which was equally leveled by the conventional therapy group at 1–3 months after baseline.

In total, five studies [61,62,63,66,70] included histopathological biopsies in their results. Taken together, collagen deposition seems to evolve more rapidly in tSVF treatment of a wound compared to controls (saline or conventional dressings). No studies with direct comparison between tSVF harvest techniques have been conducted.

Wound healing was additionally studied in five animal studies (mice, rats, rabbits), of which two used xenografts to obtain tSVF [60,80] (Table 7). Tissue SVF was isolated using the FAT procedure (*n* = 3), SVF gel (*n* = 1) and by shuffling and a single centrifugation step (*n* = 1).

Tissue SVF accelerated dermal wound healing in all ten human and five animal studies. Taken together, enzyme immunoassay results showed 7–10 times more gene expression of vascular endothelial growth factor (VEGF) and basic fibroblast growth factor (bFGF). An increase in inflammatory cells and higher capillary density with higher expression of MCP-1 and VEGF were shown^7^. An in vivo study with 15 rabbit models also showed lower expression of IL-6 and MCP-1, and lower collagen density and less alpha-SMA, myofibroblasts and COL-1, were observed post-injection with tSVF. These animal studies show that subcutaneous administration of tSVF modulates inflammation.

### 3.3. Osteoarthritis

The application of tSVF in osteoarthritis is relatively new and yet unstudied in humans. The available literature describes the usage of the Fastem and Fastkit systems [85,86], but since these procedures do not use a form of intersyringe processing, these papers were not included in the present review.

The study by Li et al. was conducted in a canine osteoarthritis model, in which tSVF was obtained by mincing inguinal fat, followed by intersyringe processing 90 times through a one-hole 2.0 mm fractionator, followed by a final centrifugation step [29]. MRI and histological analysis at 3 months FU showed complete filling of the cartilage repair after tSVF application versus poorly filled in the control group (purely minced fat graft). Gene expression of collagen type II was upregulated and collagen type I downregulated, while higher international cartilage repair society (ICRS) scoring was reported after SVF treatment vs control. The restoration of cartilage injury after tSVF administration is promising and should be studied further for future application in common arthritis manifestations, for example, thumb carpometacarpal (CMC) joint osteoarthritis.

### 3.4. Other

Four clinical studies outside the categories mentioned in this scoping review (that is skin rejuvenation, wound healing and osteoarthritis) are worth mentioning. A pilot study (*n* = 10) was performed in alopecia patients who were treated with tSVF harvested using the FAT procedure in combination with PRP. In all ten patients, hair follicle density increased, and regrowth was observed. The restoration of hair follicles may implicate the ability of tSVF and PRP to stimulate follicular regeneration through boosting resident epidermal stem cells. Hair growth was also faster in tSVF (FAT procedure)-treated mice, and in vitro analysis showed increased dermal papilla cell proliferation, migration and cell cycle progression of the injected lipoaspirate contained adipose stem cells in combination with the extracellular matrix [84].

Improvement in interstitial cystitis in six women was obtained with repetitive intravesical injections with tSVF (nanofat). Both pain—VAS and related symptoms improved significantly at 18-month follow-up after four intravesical injections. The intravesical morphology was restored based on cystoscopic examination [65]. After four injections (1 year FU) all inflammatory cells (T-lymphocytes, eosinophils and mast cells) were decreased in vulvar lichen sclerosis. Associated symptoms and quality of life also improved significantly in comparison with conventional topical clobetasol propionate cream. Glottis closure and voice quality expressed in the GRBAS scale improved significantly in 22 tSVF-treated (SVF gel) unilateral voice fold paralysis patients [73]. The effects were observed up to 18 months of follow-up. Injection of tissue SVF (MyStem) was also proposed as treatment of venous leg ulcers. In total, 58% had complete healing of the ulcer at 12 months FU, and ulcer size was decreased in 96% ± 1.7. Between the 3 month and 6 month FU, the ulcer decreased significantly (64.5%).

Volume retention of fat grafts (nanofat vs. Coleman’s fat) in mice were sampled after being exposed to shear force [82]. The retention rate of the tSVF/ECM gel was higher with a larger number of mesenchymal stem cells, supra-adventitial (SA) adipose stromal cells (ASCs), and adipose-derived stem cells but a lower number of endothelial progenitor cells. It was suggested that the pluripotency of adipose tissue-derived stem cells was increased by intersyringe processing, which can improve graft retention in fat grafting. In sixteen mice, nanofat injections showed high functional microvessel density in dermal sites after 14 days post-injection; these results suggest tSVF administration to enhance tissue vascularization.

## 4. Discussion

The generation of tSVF through fractionation of lipoaspirate is achieved by mechanical disruption of adipocytes. This results in a volume reduction after removing oil from disrupted adipocytes and their cellular remnants. Since adipose tissue consists of 90% (*v*/*v*) adipocytes on average, volume should be reduced by approximately 90% to obtain the remaining 10% (*v*/*v*) tSVF fraction. SVF gel and the FAT procedure showed the most volume reduction (approximately 90%) with an increase in SVF cells per volume unit and high viability, while mechanical micronization squeeze filtration fluid obtained tSVF with a 52% reduction in volume. Centrifugation as a first step removes fluid from the lipoaspirate, which makes adipocytes more susceptible for disruption during intersyringe shuffling and is therefore crucial for proper isolation tSVF [38]. In this review, many procedures have not published the end volume of tSVF or the quantification of the number of adipocytes to substantiate the loss of adipocytes. It is therefore impossible to conclude that these procedures result in condensed tSVF. It is likely that these procedures generated mere processed fat with an increased injectability, in other words emulsified fat, especially when centrifugation was not applied before fractionation. Filtration as a final step seems to improve injectability but does not reduce the voluminous oily layer. Moreover, part of the extracellular matrix, with its bound cells and growth factors, retain in the filter, which could lead to reduced clinical efficacy [32].

Technical improvements in these procedures are only relevant when they translate to improved clinical outcomes. Various studies have shown beneficial effects of tSVF in the field of skin rejuvenation, volume retention and wound healing. At this point, no superior isolation method for tSVF and associated clinical outcomes can be identified, since the study methods and endpoints vary widely and thus comparisons between studies are difficult to make. Volume reduction and concentration of tSVF seem especially important in medical conditions that necessitate injection in small volumes, such as small joints, scars or perianal fistulas. Hypothetically, injection under these conditions could show improved clinical outcomes when using concentrated tSVF instead of regular fat transfer. Furthermore, shear stress of fractionation activates cells, which could stimulate their regenerative potential, as was previously published [41].

In general, tSVF seems to have a more positive clinical outcome when diseases with a high proinflammatory character are treated, such as osteoarthritis or (disturbed) wound healing, in comparison with rejuvenation of the aging skin. Pathological processes result in a disbalance in extracellular factors such as inflammation, excessive ECM deposition and crosslinking, or a lack of angiogenesis. The difference in clinical outcome suggests that tSVF needs a trigger such as inflammation to ‘re-educate’ damaged tissue.

There are a few limitations of this review. The scoping instead of systematic character of this review might result in missing studies that would otherwise have been included despite our considerable efforts in our searches. In this review, only fractionation procedures based on the shuffling principle using a device with holes or blades were included. To identify the appropriate isolation method of SVF, other mechanical isolation or enzymatic isolation procedures should be included as well. On the other hand, the goal of this review is to correlate differences in techniques between shuffling procedures to clinical efficacy.

Most of the validation studies of the reported procedures do not report the complete set of validation data that is defined by the joint statement of the IFATS and ISCT. Multiple studies reported differentiation capacity and phenotype using flow cytometry, but only three procedures reported colony formation capacity. Conclusions cannot be drawn from a comparison of cell number yield because cell numbers vary between donors [20]. Also, differences in lipoaspirate handling protocols and quantification methods have a large impact on cell yield, which makes absolute comparison almost impossible [24]. We recommend that future validation studies of procedures report the complete set of validation data as defined by the joint statement of the IFATS and ISCT [9]. They should focus on providing evidence for the fractionation of stromal vascular fraction by reporting end volume, loss of adipocytes and increase in number of SVF cells.

Future studies should be well designed and include validated endpoints that are measured and reported homogeneously. It would be of interest to compare different isolation methods within a specific clinical field, preferably in a randomized fashion. As wound and scar healing is known to expand over a minimum of twelve months, we suggest monitoring follow-up results at least twelve months postoperatively. Adding therapies to tSVF treatment, such as PRP, PRF or CO_2_ laser, demand another control group in the study design, as the exact effect of tSVF is still under debate.

Standardization of procedures is of paramount importance to analyzing and comparing the results of clinical studies. Many clinical studies were found that described procedures that do not correspond with the original description and validation of the methods of these procedures. It is recommended that authors strictly adhere to the methods of procedures because otherwise additional validation of their altered procedures are necessary and clinical comparisons are ambiguous.

tSVF injection is a promising new therapy that is easy to use, can be injected during the same surgical procedure and shows positive results. In our opinion, the future of tSVF lies especially in clinical indications with a proinflammatory character such as osteoarthritis and wound healing. Future well-designed clinical trials should focus on using validated procedures with validated outcome measurement tools.

## Figures and Tables

**Table 1 bioengineering-10-01175-t001:** Overview of procedures. (Figure was created using BioRender.com, accessed at 24 July 2023).

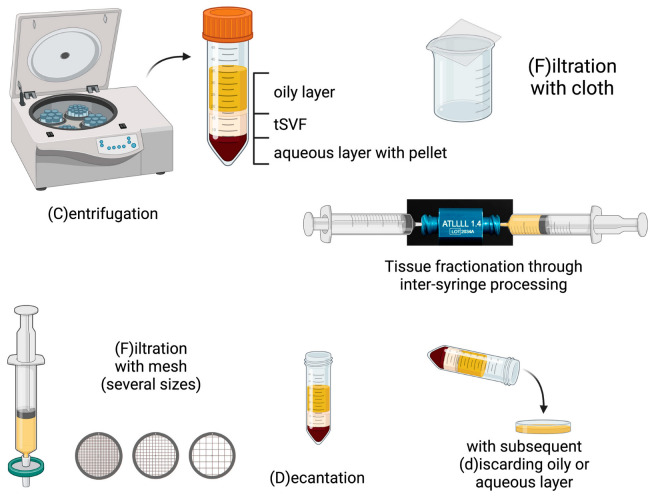			
	Step 1 Tissue Separation	Step 2 Tissue Fractionation	Step 3 Tissue Separation
**Filtration**			
Nanofat [15]	F (cloth)	One-hole fractionator	F (cloth)
Nanofat 2.0 [18]	F (cloth)	One-hole fractionator	-
Nanofat cell aggregates [19]	F (mesh)	One-hole fractionator, 2.4 mm then 1.4 mm	F (mesh)
LipocubeNANO [20]	D + F	Undefined connector	F (mesh)
**Centrifugation**			
FAT procedure [16]	C + d	Three-hole fractionator (1.4 mm)	C + d
FAT procedure 2.0 [21]	C + d	One-hole fractionator (1.4 mm)	C + d
Mechanical micronization: squeeze [22]	C + d	Slicer with spinning blade	C + d
Lipocube [17,23]	D	Through blade grid (1000–750–500 µm)	Add buffer + incubation C
Centrifuge-modified nanofat [24]	C + d	Fractionator	
Evo-modified nanofat [24]	C + d	Fractionator	
ARAT [25]	C + d	Mesh (4000–2400–1200–600–400–200–100 µm)	
MEST [25]	C + d	Mesh (4000–2400–1200–600–400–200–100 µm)	C
**Centrifugation and filtration**			
SVF gel [26]	d + C + d + collect oil	One-hole fractionator (1.4 mm)	F (mesh) Add oil C + d
Mechanical micronization: Emulsification [22]	C + d	Three-hole fractionator (1.4 mm)	C + d F (mesh)
Supercharge-modified nanofat [24]	(1) F (mesh) + C	(2) fractionator	Add (1) + (2)

**Table 2 bioengineering-10-01175-t002:** Detailed overview of processing steps of the procedures.

	Step 1 Tissue Separation	Step 2 Tissue Fractionation	Step 3 Tissue Separation	Reduction in Volume (%) *
**Filtration**				
Nanofat [15]	Filtering over nylon cloth 500 µm	30× through one-hole (undefined size) connector between 10 cc syringes	Filtering over nylon cloth 500 µm	NR
Nanofat 2.0 [18]	Filtering over nylon cloth 500 µm	30× through one-hole (undefined size) connector	-	NR
Nanofat cell aggregates [19,20]	Fluids expelled by manual pressure through filter device (Fat press, Tulip medical; undefined mesh size)	1. 30× through one-hole 2.4 mm 2. 30× through one-hole 1.2 mm between 20 cc syringes	Filtering through 600 µm to 400 µm mesh screen	15%
LipocubeNANO: [20]	Decantation 1× through undefined filter (port 1)	10× between undefined holes (port 2 and 3)	1× through a 500 µm filter (port 4)	NR
**Centrifugation**				
FAT procedure [16]	Centrifugation (3000 rpm; radius 9.5 cm; 2.5 min) Oil + liquid discarded	30× through three-hole 1.4 mm	Centrifugation (3000 rpm; radius 9.5 cm; 2.5 min) Oil+liquid discarded	90% [83–93%]
FAT procedure 2.0 [21]	Centrifugation (3000 rpm; radius 9.5 cm; 2.5 min) Oil + liquid discarded	30× through one-hole 1.4 mm	Centrifugation (3000 rpm; radius 9.5 cm; 2.5 min) Oil + liquid discarded	89 ± 4%
Mechanical micronization: squeeze [22]	Centrifugation (1200× *g* for 3 min) Oil + liquid discarded	Automated slicer with a spinning sharp blade	Centrifugation (1200× *g* for 3 min) Oil + liquid discarded	52 ± 6% **
Lipocube [17,23]	Decantation	10× through blade grid 1000 µm holes 10× through blade grid 750 µm holes 10× through blade grid 500 µm holes between 20 cc metallic pistons	Add Ca-Mg balanced buffer solution (ratio 1:3) Incubation 10 min Centrifugation (2000× *g* for 10 min) Collect the pellet and resuspend	NR
Centrifuge-modified nanofat [24]	Centrifugation (1300 rpm for 10 min) and fat is collected (pellet is discarded)	30× through Luer lock connector (undefined size)	-	NR
Evo-modified nanofat [24]	Centrifugation (80 RPM × 3 min) and fat is collected (pellet was discarded)	30× through Luer lock connector (undefined size)	-	NR
ARAT [25]	Centrifugation (500× *g* for 2 min) Lower liquid layer was discarded	Shuffling 10 mL/s between 10 cc or 20 cc syringes with a blade mesh in between: 4000 µm 2400 µm 1200 µm 600 µm 400 µm 200 µm 100 µm	-	NR
MEST [25]	Same as ARAT	Same as ARAT	Centrifugation (1200 g for 6 min) and bottom two layers are used (Stromal cell solution and stromal cell aggregate)	NR
**Centrifugation and filtration**				
SVF gel [26]	Liquid discarded Centrifugation (1200× *g* for 3 min) Liquid discarded Collect oil layer	0.5–5 min through one-hole Luer lock connector 1.4 mm *** between syringes (undefined size)	Filtering over mesh filter 500-µm Add 0.5 mL of collected oil and mix by shifting 3–5× Centrifugation (2000× *g* for 3 min) Discard oil layer	85–90%
Mechanical micronization: Emulsification [22]	Centrifugation (1200× *g* for 3 min) Oil+liquid discarded	30× through three-hole 1.4 mm Luer lock connector between 2.5 cc syringes	Centrifugation (1200× *g* for 3 min) Oil discarded Filtering over mesh filter 500 μm	(tissue in the filter) 61 ± 5% ** (fluid after the filter) 90 ± 3% **
Supercharge-modified nanofat [24]	Lipoaspirate divided in two parts **Part 1** Automatic filtration (120 µm filter) Centrifugation (1300 rpm for 10 min) Pellet was collected	**Part 2** 30× through Luer lock connector between 10 cc syringes	**Part 1** is then added to **part 2**	NR
**Other methods**				
Emulsified fat by An et al. [27]	Washed with phosphate-buffered saline (PBS) Minced with scissors	30× through three-hole 1.0mm connector between 2 cc syringes		NR
tSVF gel by Wang et al. [28]	Washed with saline Mincing with scissors	10 mL/s for 1 min between syringes (undefined size)	Centrifugation (2000× *g* for 3 min) Oil+liquid discarded	NR
ECM/tSVF gel by Li et al. [29]	Mincing with scissors	90× through one-hole 2.0 mm connector between 10 cc syringes	Centrifugation (2000× *g* for 3 min) Oil+liquid discarded	NR

±SD (range) NR = not reported. * End volume compared relatively to starting volume (start volume 1.0 and end volume is a fraction of this). ** Only recorded relative to centrifuged fat. *** In their first article, Yao et al. mention a 2.4 mm connector; however, in their follow up studies, a 1.4 mm connector is reported.

**Table 3 bioengineering-10-01175-t003:** Overview of reported validation data. (Figure was created using BioRender.com, accessed at 24 July 2023).

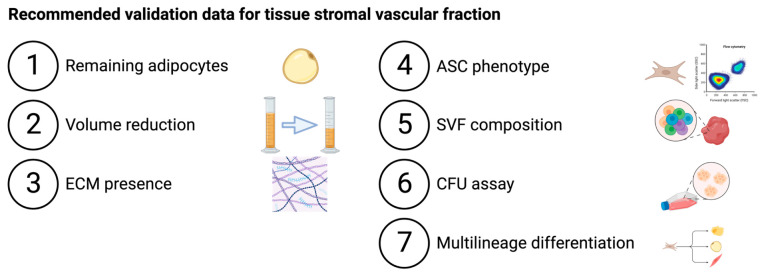							
	Adipocytes Removal		ECM Presence	SVF Composition, Cultured ASC Phenotype and Characterization			
	Adipocytes	Volume	ECM	ASC Phenotype	SVF Composition	CFU Assay	Multilineage Differentiation
**Filtration**							
**Nanofat**	**-**	**-**	**-**	**✓**	**✓**	**-**	**Adipo+**
**Nanofat 2.0**	**-**	**-**	**-**	**✓**	**-**	**-**	**-**
**Nanofat cell aggregates**	**-**	**✓**	**-**	**-**	**-**	**-**	**Adipo+**
**LipocubeNANO**	**-**	**-**	**-**	**-**	**✓**	**-**	**Adipo+**
**Centrifugation**							
**FAT procedure**	**Immunohistochemistry**	**✓**	**Histology**	**✓**	**✓** *****	**✓** *****	**Adipo+ Osteo+ SMC+**
**FAT procedure 2.0**	**Immunohistochemistry**	**✓**	**Histology**	**-**	**✓**	**✓**	**-**
**Mechanical micronization:** **s** **queeze**	**Confocal microscopy**	**✓**	**SEM**	**-**	**✓**	**-**	**-**
**Lipocube**	**-**	**-**	**-**	**✓**	**✓**	**✓**	**-**
**Centrifuge-modified nanofat**	**-**	**-**	**-**	**-**	**✓**	**-**	**-**
**Evo-modified nanofat**	**-**	**-**	**-**	**-**	**✓**	**-**	**-**
**ARAT**	**Histology**	**-**	**-**	**-**	**✓** ******	**✓** ******	**-**
**MEST**	**Histology**	**-**	**-**	**-**	**✓** ******	**✓** ******	**-**
**Centrifugation and filtration**							
**SVF gel**	**Confocal microscopy**	**✓**	**SEM**	**-**	**✓** ******	**-**	**Adipo+ Osteo+** **Chondro+**
**Mechanical micronization:** **emulsification**	**Confocal microscopy**	**✓**	**SEM**	**-**	**✓**	**-**	**-**
**Supercharge-modified nanofat**	**-**	**-**	**-**	**-**	**✓**	**-**	**-**

Legend: SEM = scanning electron microscopy. * data were reported in the FAT 2.0 paper. ** no quantification was provided.

**Table 4 bioengineering-10-01175-t004:** Tissue validation data of procedures.

	Histology/Immunohistochemistry	Viability	Other
**Filtration**			
Nanofat [15]	-	Fluorescence live/dead staining: no viable adipocytes are visible	-
Nanofat 2.0 [18]	Oil-red O staining: Showed marked damage at cellular level.	-	-
Nanofat cell aggregates [19,20]	-	-	-
LipocubeNANO: [20]	-	-	-
Centrifugation			
FAT procedure [16]	Quantified (immuno)histochemistry: Alpha-SMA: 6.2% ± 5.5 vWF 7.0% ± 4.2 Masson’s Trichrome SVF more than control (no quantification)	Fluorescence live/dead staining: 100%	-
FAT procedure one hole [21]	Quantified immunohistochemistry: Alpha-SMA: 0.83% ± 0.33 Perilipin A: devoid of adipocytes More collagen than in control	-	-
Mechanical micronization: squeeze [22]	BODIPY/Lectin/Hoechst staining: Showed structural damage compared to normal fat and exhibited occasional aggregations of capillary fragments. Perilipin/Lectin/Hoechst staining: Irregular-sized/shaped adipocytes, and lipid droplets and fragmented capillaries. The irregular-sized/shaped adipocytes seemed to be damaged or dead, although they still expressed perilipin.	-	Scanning electron microscopy: Damaged adipocytes.
Lipocube [17,23]	-	-	-
Centrifuge-modified nanofat [24]	-	-	-
Evo-modified nanofat [24]	-	-	-
ARAT/MEST [25]	HE staining after 4000 micron, 2400 micron, 1200 micron, 600 micron, 400 micron and 200 micron adinizing: Intact adipocytes could be seen after all adinizing sessions.	-	Secretome after explant culture: (pg/mL): VEGF-A: 43.52 ± 12.21 EGF-A: 16.44 ± 2.67 FGF-2 8519.31 ± 3122.42 PDGF: 64.60 ± 21.43 NGF: 26.12 ± 14.78 TGFB1: 840.94 ± 115.77 Anti-inflammatory IL-10: 246.77 ± 116.86 IL-1ra: 417.21 ± 211.37 Inflammatory IFNg: 2.20 ± 1.85 IL-1b: 1221.44 ± 664.37 IL-6: 17,338.21 ± 3224.60 TNFa: 68.12 ± 21.44
Centrifugation and filtration			
SVF gel [26]	Confocal Microscopy using Hoechst(blue)/Lectin(red)/BODIPY(yellow): After mechanical process multiple lipid droplets. After centrifugation most of the free lipid drops were discarded leaving very few flat, fragmented adipocytes. Also, density of vessel-associated connective tissue increased.	-	Scanning electron microscopy: Level of fragmentation of ECM increased with the duration of processing time (*p* < 0.05 for 5 min processing).
Mechanical micronization: Emulsification [22]	BODIPY/Lectin/Hoechst staining: Showed structural damage compared to normal fat and exhibited occasional aggregations of capillary fragments. Perilipin/Lectin/Hoechst staining: Irregular-sized/shaped adipocytes, and lipid droplets and fragmented capillaries. The irregular-sized/shaped adipocytes seemed to be damaged or dead, although they still expressed perilipin.	-	Scanning electron microscopy: Filtrated fluid of emulsified fat showed more damaged adipocytes compared to centrifuged fat. Filtrated fluid of emulsified fat was filled with extracellular matrix fragments and contained few intact adipocytes.
Supercharge-modified nanofat [24]	-	-	-
**Other methods**			
Emulsified fat by An et al. [27]	-	-	-
tSVF gel by Wang et al. [28]	-	-	-
ECM/tSVF gel by Li et al. [29]	-	-	-

±SD (range).

**Table 5 bioengineering-10-01175-t005:** Validation data of procedures after digestion with collagenase.

	Viability	Cell Number	Culture/Differentiation	Colony Formation Assay	ASC Phenotype	SVF Composition	Other
Filtration							
Nanofat [15]	Fluorescence microscopy: Only very few dead cells.	1.975 × 10^6^/100 mL lipoaspirate CD34+ 1 × 10^5^/100 mL	Adipogenic+	-	CD44 96.32% ± 1.32 CD90 67.38% ± 2.45 CD105 82.65% ± 2.07 CD14 6.67% ± 1.99 CD34 20.01% ± 1.63 CD45 13.68% ± 2.09 *	CD34+ 0.1 to 0.2 × 10^6^ cells/100 mL lipoaspirate (4.5–6.5%)	-
Nanofat 2.0 [18]	-	-	-	-	CD44 98.88% ± 0.71 CD90 65.58% ± 2.95 CD105 75.83% ± 2.88 CD14 5.45 ± 1.87 CD34 14.86% ± 2.09 CD45 3.45% ± 2.72	-	MTT assay: No difference between microfat/nanofat/nanofat 2.0
Nanofat cell aggregates [19,20]	Image cytometry: 76.80% - Muse flow cytometer: 96.05%	DNA quantification: 7.3 million cells/g Cell yield: 6.63 million cells/g lipoaspirate Muse flow cytometer: 1.44 × 10^6^/cc	Adipogenic+	-	-	-	Gene expression profiles (mRNA levels): PPAR2 7.1-fold higher than NanoTransfer Adiponectin 1.7-fold higher than NanoTransfer
LipocubeNANO: [20]	Muse flow cytometer: 96.75%	Muse flow cytometer: 2.24 × 10^6^/mL	Adipogenic+	-	-	CD45− CD90+ 7.92% CD73+, CD90+ 37.29% CD45− CD31+ 11.99% CD45+ CD14+ 2.43% CD13+ 42.04% CD73+ 53.5% (extracted from figure) CD90+ 55.82% CD146+ 53.2% CD34+ 18.84%	-
Centrifugation							
FAT procedure [16]	-	Bürker Turk counting chamber: 2.7 × 10^6^ ± 1.1 cells/mL	Adipogenic+ Osteogenic+ Smooth muscle cell+	1.29% ± 0.045 *	CD90 CD31− CD45− 99.8% ± 0.2; CD29 99.8% ± 0.2 CD44 99.0% ± 0.7 CD105 95.9% ± 4.5	CD45− CD90+ CD105+ 41.4% ± 16.5% CD31+ CD34+ 12.0% ± 4.5% CD45+ CD34− 5.3% ± 3.6% CD34+/− CD31−CD146+ 0.3% ± 0.3% CD45+ CD34+ 0.1 ± 0.2% CD34 bright CD31− CD146− uncountably low *	-
FAT procedure one hole [21]	-	Bürker Turk counting chamber: 2.35 × 10^6^ ± 0.30	-	1.29% ± 0.038	-	CD45− CD90+ CD105+ 44.9% ± 18.2% CD31+ CD34+ 19.1% ± 2.3% CD45+ CD34− 5.3% ± 3.6% CD34+/− CD31− CD146+ 0.5% ± 0.5% CD45+ CD34+ 0.2% ± 0.3% CD34 bright CD31− CD146− uncountably low	-
Mechanical micronization: squeeze [22]	Nucleocounter NC-100: 89.9 ± 4.6 percent	-	Culture: After 1 week normalized number of cultured adipose-derived stromal cells 1.1 ± 0.1 × 10^6^	-	-	Number of cells/1mL of fat CD45− CD31− CD34+ 1.5 × 10^5^ CD45− CD31+ CD34− 1.1 × 10^5^	Mass fraction: 75% ECM 25% adipocytes
Lipocube [17,23]	85.82% ± 5.74% 97.55% **	1.34 × 10^6^/mL ±1.69 0.94 × 10^6^ **	-	Cell proliferation test: On day 7, more cluster formation in mechanically digested SVF. A_490_ value after 72 h 201 ± 0.1 (*p* ≤ 0.05)	CD90 11.39% CD44 21.45% CD105 9.0% CD73 6.16%	CD45− CD73+ 42.37% CD90+ CD73+ 52.08% CD45− CD31+ 21.06% CD45+ CD14+ 7.28%	Gene expression analysis: PPAR2/adiponectin 1.43- and 1.32-fold higher in mechanically digested SVF; Col1A PCR test: 12.5 [1.2] × 10^6^; *p* ≤ 0.05, (1.5-fold higher compared to enzymatically digested SVF)
Centrifuge-modified nanofat [24]	-	Hemocytometer: 53,334 ± 8000 nucleated cells/mL	-	-	-	CD44 65.6% ± 9.1% CD90 59.7% ± 6.9% CD73 27.2% ± 4.8% CD34 55.3% ± 7.5% CD31 21.5% ± 5.1% CD146 19.7% ± 3.9%	-
Evo-modified nanofat [24]	-	Hemocytometer: 125,000 ± 12,000 nucleated cells/mL	-	-	-	CD44 68.3% ± 7.8% CD90 56.9% ± 6.1% CD73 32.3% ± 5.5% CD34 50.4% ± 5.2% CD31 17.8% ± 4.8% CD146 20.7% ± 5.1%	-
ARAT/MEST [25]	Flow cytometry: IP 1 94% ± 2 IP 2 93% ± 2 IP 3 93% ± 2 IP 4 91% ± 4 ***	Flow cytometry: IP 1 0.91 10^6^/mL IP 2 0.88 10^6^/mL IP 3 1.61 10^6^/mL IP 4 1.47 10^6^/mL ***	-	CFU-F assay: Results not reported	-	Quantification not reported	-
Centrifugation and filtration							
SVF gel [26]	-	0.5-min SVF gel: 2.7 ± 0.3 × 10^5^ cells/mL 1 min SVF gel: 4.1 ± 0.3 × 10^5^ cells/ml	Adipogenic+ Osteogenic+ Chondrogenic+	-	-	Quantification not reported, only figure.	-
Mechanical micronization: emulsification [22]	Nucleocounter NC-100: residual tissue 90.6% ± 2.8%, filtrated fluid 39.3% ± 9.1%	-	After 1 week of culture, number of cells in residual tissue of emulsified fat (5.1 ± 0.7 × 10^5^)	-	-	Number of cells/1 mL of fat CD45− CD31− CD34+ 1.4 × 10^5^ CD45− CD31+ CD34− 0.8 × 10^5^	-
Supercharge-modified nanofat [24]	-	Hemocytometer: 200,000 ± 15,000 nucleated cells/mL	-	-	-	CD44 71.2% ± 8.0% CD90 62.8% ± 7.2% CD73 30.1% ± 5.4% CD34 58.1% ± 6.3% CD31 19.9% ± 4.4% CD146 22.1% ± 4.5%	-
Other methods							
Emulsified fat by An et al. [27]	Trypan blue staining: 58.2%	Nucleocounter NC-100: 4.53 × 10^6^	-	-	-	CD45− CD34+ 12.40% ± 0.86%	-
tSVF gel by Wang et al. [28]	7-AAD staining: 80%	-	-	-	-	CD34+ CD31− CD45− 64% CD34+ CD31+ CD45− 28%	-
ECM/tSVF gel by Li et al. [29]	-	-	Chondrogenic+ Osteogenic+ Adipogenic+	-	CD29+ 97.3% CD90+ 98.7% CD105+ 99.7% CD34+ 1.3% CD45+ 1.2%	-	-

* Quantification was derived from another manuscript describing these data. ** Two studies reported validation data of the same procedure. *** Results were stratified based on reported indication protocols (IP).

**Table 6 bioengineering-10-01175-t006:** Overview of human clinical results in the literature.

	Study	Year	Study Design	N	Max. FU (m)	Categorized Method ^%^	Additional Product	Clinical Endpoints	Clinical Results * (Index vs. Control)	Overall Result
**Skin and volume enhancement**										
**1**	van Dongen et al. (2021) [45]	2016–2019	RCT	28	12	FAT procedure	PRP + tSVF vs. PRP + saline	VISIA, FACE-Q, complications	No superior result in skin quality or satisfaction. No major complications	+/−
**2**	Zhang et al. (2022) [46]	2018–2020	RCT	63 (34 vs. 29)	12	FAT procedure ^%^	tSVF vs. Coleman’s	Volume ratio on 3D imaging, retention rate, satisfaction 5p Likert	Increased contour ratio 0.87 ± 0.02 to 0.89 ± 0.03 *, higher retention * 41.2 ± 8.4% vs. 32.6 ± 8.8%, higher satisfaction (4/5p) 79% vs. 62% *	+
**3**	Cai et al. (2019) [47]	?	P	50 (28 vs. 22)	12	FAT procedure ^%^ (with 2.4 mm fractionator)	tSVF vs. BTXa	Global Aesthetic Improvement Scale (GAIS), patient satisfaction, histological analysis (*n* = 1)	Higher GIAS and satisfaction in high-grade wrinkles group *, increased collagen density	+
**4**	Wang et al. (2021) [48]	2017–2019	P	18 (6 vs. 12)	12	FAT procedure ^%^	tSVF vs. PBS	Ultrasonogram, volume measurement	Similar elasticity, increased volume of 2.3 ± 0.3 mL	+/−
**5**	Ding et al. (2023) [49]	2020–2021	P	31	15	FAT procedure ^%^	tSVF	Survival, volume, GAIS Reoperation rate	65.3% ± 6.1 2.2 ± 0.8 12.9%	+/−
**6**	Xia et al. (2022) [50]	2017–2021	P	33 (66 temples)	6	FAT procedure ^%^	tSVF	Hollowness Severity Rating Scale, satisfaction 3p Likert	91% absent hollowness, 94% satisfied	+
**7**	Luo et al. (2020) [51]	2017–2018	P	33 (66 eyes)	13	FAT procedure ^%^	tSVF	GAIS, depth measurement, retention rate	2.5 [0.5] Improvement in all depth measurements 73 ± 10%	+
**8**	Zhao et al. (2021) [52]	2018	P	18	12	FAT procedure, 1.4 mm ^%^	tSVF	Number of inflammatory lesions, Investors Global Assessment scale, biopsies at 1-month FU	Decrease in lesions, 7.3(2.7) vs. 0.7(0.7) *, 2.5(0.5) vs. 0.6(0.5) *, decrease in CD4+ T cells after 4 weeks	+
**9**	Cao et al. (2022) [53]	2017–2019	P	13	10	FAT procedure ^%^	tSVF	Satisfaction	Improvement 84 ± 3 vs. 31 ± 3 *	+
**10**	Liang et al. (2018) [54]	2014–2016	P	231 (103 vs. 128)	24	Nanofat	tSVF + PRF vs. hyaluronic acid	VISIA, SOFT5.5, satisfaction	Facial skin texture improved in both groups *, higher satisfaction rate	+
**11**	Wei et al. (2017) [55]	2014–2016	P	139 (62 vs 77)	24	Nanofat	tSVF + PRF vs. Coleman’s procedure	VISIA, SOFT5.5, satisfaction	Skin quality improvement *, higher satisfaction 90% vs. 70% *	+
**12**	Menkes et al. (2020) [56]	2018	P	50	6	Nanofat	Nanofat + PRP	Satisfaction (10p Likert), biopsies	Improvement in texture, elasticity, glow Increase in collagen and elastic fibers *, higher cellularity and vascular density *	+
**13**	Menkes et al. (2021) [57]	2017–2018	P	50	18	Nanofat	Nanofat + PRP	Vaginal health index, Female Sexual Distress Scale Revised	Improvement in VHI, 9.2 ± 1.7 vs. 3.4 ± 1.5 *, FSDS-R, 3.4±3.7 vs. 32.9±9.5*	+
**14**	Uyulmaz et al. (2018) [58]	2013–2016	R	52	5	Nanofat	-	Photographs, satisfaction (yes/no)	Improvement in skin appearance 93%, satisfactory result 18% rater vs. 92% patient	+
**15**	Zhu et al. (2022) [59]	2016–2020	R	103 (58 vs. 48)	9	FAT procedure ^%^	tSVF (FAT ^%^) vs. tSVF (nanofat)	Clinical data, satisfaction	Comparable improvement Less reoperations in FAT-treated patients*, higher satisfaction in FAT-treated patients *	+
**16**	Yao Yao et al. (2018) [60]	2015–2017	R	204 (126 vs. 78)	11	tSVF gel	tSVF gel vs. Coleman’s procedure	Photo analysis, satisfaction (5p Likert scale), histological analysis (*n* = 1)	Higher Likert tSVF *, lower rate of 2nd surgery tSVF 10.9% (11/101) vs. 32.1% (25/78) * No cysts, fibrosis or calcification	+
**Wound healing**										
**17**	van Dongen et al. (2022) [61]	2016–2020	RCT	40 (20 vs. 20)	12	FAT procedure	tSVF vs. saline	Histological biopsies, POSAS, photographs, blinded analysis with VAS	Equal collagen alignment, depth, width, POSAS patient 14.4 ± 7.6 vs. 15.3 ± 9.0, observer 14.5 ± 6.4 vs. 14.6 ± 8.8, no differences in VAS scores, low agreement	+/−
**18**	Abouzaid et al. (2022) [62]	2019–2020	RCT	100 (50 vs. 50)	3	Centrifuged-modified nanofat ^%^	Coleman’s fat + tSVF vs. conventional dressings	Photo + clinical analysis, biopsies	Decrease in hospital stay and reoperation *, less contractures *, rapid collagen deposition *	+
**19**	Gu et al. (2018) [63]	2014–2016	P	20	6	FAT procedure ^%^	-	POSAS and photographs, biopsies	Patient total 28.8 (1.0) vs. 12.2 (0.8) *, observer total 18.0 (0.7) vs. 9.2 (0.4) *, melanin AOD basal cell layer 0.8 (0.1) vs. 0.7 (0.1), no difference in elastic fibers	+
**20**	Bhooshan et al. (2019) [64]	2015–2016	P	34	3	Nanofat (without first filtration step)	-	POSAS and photographs, aesthetic result (total POSAS score)	Patient total 14 ± 4.4 vs. 27.4 ± 7.5 *, observer total 18 ± 6.8 vs. 31 ± 8.5 *, 77% good outcome	+
**21**	Hung et al. (2022) [65]	2019	P	6	6	Nanofat	Nanofat + PRP	Pain-VAS, PROM, cystoscopy	Improvement in PROMs *, 100% remission of lesions	+
**22**	Rageh et al. (2021) [66]	?	P	30	6	Nanofat		Vancouver scar scale, biopsies	Lower VSS scores in height and pliability *, improved epidermal thickness *, increased collagen (52%) and elastic fibers (50%) *	+
**23**	Huang et al. (2021) [67]	2017–2020	R	44	12	Nanofat	-	FACE-Q, assessment of photographs	Overall satisfied, 30% complete healing, 41% obvious improvement, 9% no effect	+/−
**24**	Cantarella et al. (2019) [68]	?	Pi	7	6	Centrifuged- modified nanofat ^%^	-	Videolaryngo- stroboscopy, max phonation time, VHI, EAT10	Improvement in glottic closure, longer phonation *, reduction in VHI *, improved swallowing	+
**25**	Tenna et al. (2017) [69]	2014–2015	P	30 (15 vs. 15)	12	Centrifuged- modified nanofat ^%^	tSVF + PRP vs. tSVF + PRP + CO_2_ laser	Ultrasound, FACE-Q	Improvement in subcutaneous tissue of 0.67 vs. 0.63 cm, comparable FACE-Q	+
**26**	Deng et al. (2018) [70]	2016–2017	P	20 (10 vs. 10)	0.5	SVF gel	tSVF vs. control	Wound healing rate, biopsies	35 ± 11% vs. 10 ± 3% *, decreased lymphocyte infiltration, more * and thicker collagen deposition, more new vessels *	+
**Other indications**										
**27**	Stevens et al. (2018) [71]	2016–2016	P	10	3	FAT procedure	tSVF + PRP	Hair density	30.7 hairs/cm^2^ (range 5–59), regrowth observed	+
**28**	Gutierrez et al. (2022) [72]	2017–2019	Pi	19 (9 vs. 10)	12	SVF gel	tSVF + PRP vs steroid injection	Skin elasticity, VAS, QoL (Skindex-29), biopsies	No elasticity improvement. Improvement in symptoms, but not in pain *. Improvement in QoL *, 79.7 ± 33.2 to 59.7 ± 24.9. Decrease in all inflammatory cells *.	+
**29**	Sun et al. (2021) [73]	2017–2018	P	22	18	SVF gel	tSVF	Glottis closure, GRBAS voice quality	Improvement in vocal cord shape and closure, improvement in 19/22 patients *	+

Abbreviations: FU: follow-up (months); ^%^: altered categorized mechanical method by the authors; *: statistically significant, *p* < 0.05; RCT: randomized controlled trial; P: prospective; R: retrospective; Pi: pilot; FAT: fractionation of adipose tissue; PRP: platelet-rich plasma; tSVF: tissue stromal vascular fraction PRF: platelet-rich fibrin; PBS: phosphate-buffered saline; POSAS: patient and observer scar assessment scale; VAS: visual analogue scale; GRBAS: grade of dysphonia, roughness, breathiness, authenticity, strain; VHI: Voice Handicap Index questionnaire; EAT10: Eating Assessment Tool; AOFAS: American Orthopedic Foot and Ankle Society Ankle–Hindfoot Score; VISA-A: Victorian Institute of Sport Assessment—Achilles; SF-36: Short-Form Health Survey; MIDAS: Migraine Disability Assessment Score; PGIC: Patient Global Impression of Change.

**Table 7 bioengineering-10-01175-t007:** Overview of animal clinical results in the literature.

	Study	Year	Study Design	N	Max. FU (m)	Categorized Method ^%^	Additional Product	Clinical Endpoints	Clinical Results *	Overall Result
**Skin and volume enhancement**										
**1**	Akgul et al. (2018) [74]	2013–2015	P	14 rats/6 XG	1.5	FAT procedure ^%^	tSVF enriched with adipocyte fragments vs. controls	Histological biopsies	Viable adipocyte architecture, collagen accumulation, CD68 + CD44+	+
**2**	Zhu et al. (2021) [75]	NR	P	60 mice (15 vs. 45/17 XG)	3	FAT procedure ^%^	tSVF vs. control	Histological biopsies	Dermal thickness * 184.4 ± 2.8, higher collagen deposition, increased TGF-b1 and Smad 2 expression *, lower MMP2/9 *, more fibroblasts	+
**3**	Yu et al. (2018) [76]	2017	P	30 (20 vs. 10) mice/5 XG	3	Nanofat	tSVF vs. control	Histological biopsies, integrity, cysts/vacuoles, fibrosis, inflammation, capillary density (CD3+ vessels)	Better survival and morphological integrity, 3.6 ± 0.5 vs. 2.7 ± 0.9 *, 2.6 ± 0.7 vs. 3.2 ± 0.8 *, 2.1 ± 0.6 vs. 2.9 ± 0.8 *, 2.1 ± 0.6 vs. 2.6 ± 0.5 *, 24.6 ± 4.7 vs. 10.4 ± 2.9 *	+
**4**	An et al. (2020) [27]	NR	P	24 rats (16 vs. 8)	1	Other ^%^	tSVF vs. control	Histological biopsy, collagen AOD, anti-PCNA (cell proliferation), anti-CD31 (vascularization degree)	Higher AOD in 1mL SVF application *, 102 ± 12 vs. 55 ± 8 *, 95 ± 4.3 vs. 63 ± 2.7/mm^2^ *	+
**5**	VinayKumar et al. (2022) [77]	2018	P	9 guinea-pigs (9 vs. 9)	6	Nanofat	tSVF vs. control	Histological biopsies, polarized light microscopy	Similar inflammatory infiltrate and collagen fiber orientation, increase in collagen distribution *	+/−
**6**	Xu et al. (2018) [78]	NR	P	18 mice (6 vs. 12, 10 XG)	2	Nanofat	tSVF vs control and enzymatic tSVF	Histological biopsies	Increased dermal thickness *, higher capillary density and epidermal proliferation index *; high VEGF, EFG, bFGF, IGF and IL-6 *	+
**7**	Liu et al. (2021) [79]	NR	P	12 rabbits	1.5	SVF gel	tSVF and PRF vs. tSVF	Histological biopsies	Slightly more volume combined with PRF, larger adipocytes and more ordered fibroblast distribution	+/−
**Wound healing**										
**8**	Zhang et al. (2017) [80]	NR	P	10 mice /NR XG	0.5	FAT procedure ^%^	tSVF vs. control	Photographs, necrosis rate, histological biopsies	Thicker fatty layer, 22.1 ± 0.1 vs. 53.8 ± 0.1% *, 7/10× more VEGF and bFGF *, more human-derived vessels *, 43% more CD31+ vasculature *	+
**9**	Chen et al. (2019) [81]	NR	P	10 rats	0.5	FAT procedure ^%^	tSVF vs. control	Wound healing, biopsies	Faster and complete wound healing *, more vesicular structures and inflammatory cells, higher capillary density, MCP-1 and VEGF *	+
**10**	Sun et al. (2017) [7]	NR	P	54 (18 vs. 36)	0.5	FAT procedure ^%^	tSVF vs. control	Wound healing, capillary density, inflammatory reaction	Complete healing at 14 days FU, more vascularization *, sharp increase and later decrease in inflammatory cells	+
**11**	Yao Yao et al. (2016) [26]	NR	P	52 mice/17 XG	0.5	SVF gel	tSVF vs. control	Photographs	Wound healing and closure	+
**12**	Wang et al. (2019) [28]	NR	P	15 rabbits	3	Other ^%^	tSVF gel	Size, color, texture, dermal thickness, histological biopsies	Improvement, 0.5 ± 0.3 vs. 1.4 ± 0.3 mm *, CD206+ macrophages dermal layer, lower IL-6 and MCP-1 *, lower collagen density *, less alpha-SMA, myofibroblasts and COL-1 *	+
**Osteoarthritis**										
**13**	Li et al. (2020) [29]	2019	P	30 rabbits	3	Other ^%^	tSVF vs. control	Radiology (MRI), histology, immunohistochemistry, total histological outcome score, ICRS	Cartilage repair, Filled defect, strong glycosaminoglycan staining, COL-II up, COL-I down, 10.2 ± 0.8 vs. 8.4 ± 1.1 *, 9.8 ± 1.3 vs. 7.4 ± 1.1 *	+
**Other indications**										
**14**	Ye et al. (2021) [82]	NR	P	50 (25 vs. 25)/7 XG	2	Nanofat	tSVF vs. Coleman’s	Histological biopsies	Perilipin + cell density *	+
**15**	Weinzierl et al. (2022) [83]	NR	P	16 mice	0.5	Nanofat		Histological biopsies	High functional microvessel density	+
**16**	Li et al. (2020) [84]	NR	P	6 mice (6 XG)	0.5	FAT procedure ^%^	tSVF vs. decellularized tSVF	Hair growth Biopsies	Increased hair growth, increased proliferation, migration, cell cycle progression	+

Abbreviations: FU: follow-up (months); NR: not reported; ^%^: altered categorized mechanical method by the authors; *: statistically significant, *p* < 0.05; XG: xenograft; RCT: randomized controlled trial; P: prospective; R: retrospective; Pi: pilot. FAT: fractionation of adipose tissue; tSVF: tissue stromal vascular fraction; AOD: average optimal density; PCNA: proliferating cell nuclear antigen; VEGF: vascular endothelial growth factor; bFGF: basic fibroblast growth factor; ICRS: international cartilage repair society.

## Data Availability

Not applicable.

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
