# Peer review of "Mechanical Fractionation of Adipose Tissue—A Scoping Review of Procedures to Obtain Stromal Vascular Fraction"

_bioengineering, 2023, doi:10.3390/bioengineering10101175_

Round 1
Reviewer 1 Report
General Comments
This is a review manuscript outlining and commenting on the various procedures to obtain the stromal vascular fraction of adipose tissue for clinical applications.
Instead of using abbreviations such as e.g. and i.e. it would be better to use English. e.g = such as and i.e. = that is.
Specific Comments
Procedures
The text describing procedures needs to be well illustrated. The figure set at the end of the document is not sufficient.
Masson’s Trichrome staining This is not an immunological procedure.
Perilipin is present is present in early stages of adipocyte development as it surrounds lipid deposits. Adipogenesis is a constant process, as such early stage adipocytes are possible in any preparation.
et al. why not in italics?
There are minor issues with punctuation and excessive use of unnecessary abbreviations such as e.g.
Reviewer 2 Report
The review by Schipper et al. titled "Mechanical fractionation of adipose tissue, a scoping review of procedures to obtain stromal vascular fraction" is a relatively well written work regarding tSVF. The work summarizes the field well and the tables assist the reader to gain a deeper understanding of the topic.
Some English language editing needed. Mainly style and sentence structure in couple places.
Reviewer 3 Report
This manuscript is a scoping review that focuses on the mechanical fractionation of adipose tissue to obtain stromal vascular fraction (SVF). The aim of the review is to assess the efficacy of mechanical fractionation procedures to obtain tissue SVF (tSVF) and to provide an overview of the clinical efficacy of tSVF in skin rejuvenation, wound healing, and osteoarthritis. The procedures for obtaining tSVF are categorized into filtration, centrifugation, both filtration and centrifugation, and other methods. The review includes both animal and human studies that report the results after tSVF injection.
Adipose tissue contains adult mesenchymal stem cells that may modulate the metabolism when applied to other tissues. The heterogeneous pool of cells found in the SVF of adipose tissue and the purified mesenchymal stromal/stem cells (ASCs) isolated from this pool have increasingly been used as therapeutic tools in regenerative medicine. SVF can be isolated from adipose tissue mechanically and/or enzymatically.
Overall, the authors' summary is comprehensive and the paragraphs are better designed. However, in order to be more in line with the theme of the special issue, the authors should look forward to the future application of tSVF in the field of regenerative medicine and put forward their own viewpoints, rather than just summarizing the previous work. In this regard, the authors should increase the content appropriately. The numbering of the first two sections is missing (1, 2). The English needs to be improved to a certain extent. There are some errors in grammar and format in the whole manuscript: inconsistencies; tense; spelling mistakes; single and plural expressions; the use of prepositions and definite/indefinite articles; punctuation. The definite article “the” and the indefinite article “a” are missing from a lot of sentences.
The English needs to be improved to a certain extent. There are some errors in grammar and format in the whole manuscript: inconsistencies; tense; spelling mistakes; single and plural expressions; the use of prepositions and definite/indefinite articles; punctuation. The definite article “the” and the indefinite article “a” are missing from a lot of sentences.
Round 2
Reviewer 1 Report
I still believe that more figures are needed, but this is a minor comment. Otherwise, suitable for publication.
Text needs to be proofread better.
